# European Flat Oyster (*Ostrea Edulis* L.) in the Eastern Baltic as Evidence of Long-Distance Trade in Medieval and Early Modern Times

**Lembi Lõugas** [1,*] , **Inna Jürjo** [2] **and Erki Russow** [1]

1   Archaeological Research Collection, Tallinn University, 10 Rüütli Street, EE10130 Tallinn, Estonia;
    erki.russow@tlu.ee
2   Institute of History, Archaeology and Art History, Tallinn University, 5 Uus-Sadama Street,
    EE10120 Tallinn, Estonia; inna.jurjo@tlu.ee
*   Correspondence: lembi.lougas@tlu.ee

**Abstract:** Along most of the European littoral, oysters were appreciated as a wholesome and palatable food from the Stone Age onwards, yet were transported much further from their natural habitats when long-distance trade in marine foodstuffs began in medieval times. The brackish waters of the Baltic Sea are not considered a suitable environment for this mollusc, and therefore all archaeological oyster shell finds are the result of import to the eastern Baltic. In this study, over 1000 shells found in different medieval and early modern archaeological contexts in Estonia were analysed, and the obtained data recorded in a data repository. Some conclusions are set out, based on shell size and shape, and breakage traces, but more detailed taphonomic studies are left for the future. This study identifies the earliest imports of oysters recorded by archaeological material and written sources. Both show records not much earlier than the 16th century AD. Although no information is preserved about the exact origin of oysters imported to Estonia, the oyster beds most probably exploited are those in the central eastern North Sea, i.e., the Wadden Sea.

**Keywords:** oysters; eastern Baltic Sea; zooarchaeology; archaeomalacology; written sources; archaeological finds; medieval; early modern

## 1. Introduction

Zooarchaeology is usually considered to be the study of animal bones found on archaeological sites. However, dead animals can leave behind many types of remains other than bones. This is particularly the case for invertebrates, some of which have a very hard and compact exoskeleton. One of the enduring kinds of remains, if well preserved, is the mollusc shell. Depending on the speed of deposition and the soil type, shells may be preserved very well or, on the contrary, may break quite easily. Direct disposal and immediate burial results in better preservation than first keeping remains in the open air and burying them later.

There are many examples of mollusc shells recovered from archaeological material, and most of these relate to shellfish ("shellfish" being a common name used for members of several families of bivalve molluscs). Campbell [1] emphasizes the need to collect mollusc shells from archaeological sites, especially those of marine origin. Since marine molluscs do not occur in terrestrial habitats, the presence of their shells on archaeological sites is the result of human activity (not to be confused with the presence of shells in natural coastal sediments). Humans have gathered shellfish for many purposes, but mostly for food. Shells can be informative regarding subsistence, annual scheduling, the resources and habitats exploited, the effects of that exploitation on wild populations and exchange networks, the speed and efficiency of transportation, etc. [1].

For example, in Denmark, a special kind of accumulation of waste from human activity, called a *køkkenmødding* ("kitchen midden"), consists largely of mussel shells, indicating the

heavy exploitation of mussels for human food, especially during the time of the Ertebølle Culture. Accordingly, the name "shell midden" is also used to refer to these ancient dumps [2–4]. In Europe, shell middens can be found mostly along the Atlantic seaboard, but also at lakeshores, and date from the 5th–4th millennium BCE [5]. Younger middens are also found, although in the eastern Baltic only one site—Riṇṇukalns in Latvia—has yielded a typical shell midden from the 4th–3rd millennium BCE [6]. This accumulation contains only freshwater molluscs, which are available in the local Lake Burtnieki water system.

Another type of shell, often found in Iron Age and medieval find complexes, comes from the marine snail called the cowrie (*Cypraea* (*Monetaria*) *moneta*). This mollusc is most abundant in the Indian Ocean, and its shell was used worldwide as currency. It reached the eastern Baltic first by trade in the Viking Age, approximately the 7th–10th century, and there are numerous cowrie finds from burial sites of the 13th–15th century [7] (p. 149), [8] (p. 108).

None of the prehistoric assemblages from the eastern Baltic contain oyster shells; these appear only from medieval times onwards. European flat oyster (*Ostrea edulis* Linnaeus) is a native oyster species of western European coastal waters. A few centuries ago, oysters populated the shallow, as well as deeper, offshore waters of the eastern Atlantic in great numbers. Particularly abundant beds of flat oysters occurred in the central part of the North Sea, including the Wadden Sea, but from the end of the 19th century, the oyster populations suffered from heavy overexploitation [9,10]. Oysters were also cultivated, a practice that is attested in Italy as far back as the 1st century AD and developed elsewhere as well, especially in later times, after the wild populations had diminished due to overexploitation [11,12]. As the conditions in the Baltic Sea (including excessively low salinity and large temperature fluctuations in the brackish coastal zone) are not suitable for oysters [12], they cannot live in this sea; although, there have been some unsuccessful attempts to introduce oysters into the southern Baltic [13] (p. 62). Therefore, all finds of oyster shells in the eastern Baltic can be considered imports.

Besides the finds of oyster shells, other imported shellfish have also been identified in Estonian archaeological material. Not numerous, although still represented, the blue mussel (*Mytilus edulis*) and the scallop (great Atlantic scallop, *Pecten maximus*, and/or the Mediterranean Jacob's scallop, *Pecten jacobaeus*) have been recorded. Although the scallop finds are usually associated with the pilgrimage to Santiago de Compostela in north-western Spain, and in most cases the association is correct, some of the recovered fragments seem to be ordinary food waste. However, in view of the low number of such fragments in the archaeological material and the most typical find contexts (ecclesiastical sites, burial sites), culinary purpose is usually excluded. A few specimens of cockle (*Cerastoderma* sp.), whelk (*Buccinum undatum*), limpet (*Patella vulgata*) and clam (Veneridae) shells have also been found, all of them considered as imports to the eastern Baltic [14] (see Supplementary Materials). Only the shells of the freshwater mussels living in rivers and lakes are considered to be of local origin.

In Germany, on the coast of the Wadden Sea, the marketing of oysters began in the 13th century AD, when the North Frisians brought fish and oysters by boat to the market in Hamburg [15]. However, there are no indications that oysters were traded or transported inland at that time [13] (p. 60). In England, marketing to, for example, the west coast of Schleswig, Germany began in the 11th century (*ibid*.), although local marketing had already started in southern Britain during Roman times [16]. An organized German oyster fishery first developed off the islands of Sylt and Föhr on the coast of the Wadden Sea in the 16th century, still under Danish overlordship, and became an important economic activity in the 17th century. An interesting fact is mentioned by Hansen [15]: in the 17th century, Swedish merchant ships repeatedly robbed North Frisian fishermen of their catch when sailing to market in Hamburg. Hamburg was an important trading centre at that time, and most of the catch was marketed there, oysters sometimes being shipped as far as Hungary and Russia [13] (pp. 61–62). While the trade to Hungary required some transport over land, north-eastern Russia, especially St Petersburg, was accessible via the Baltic Sea. The

trade to the eastern Baltic also involved the harbours in Estonia, Latvia and Lithuania (see Figure 1).

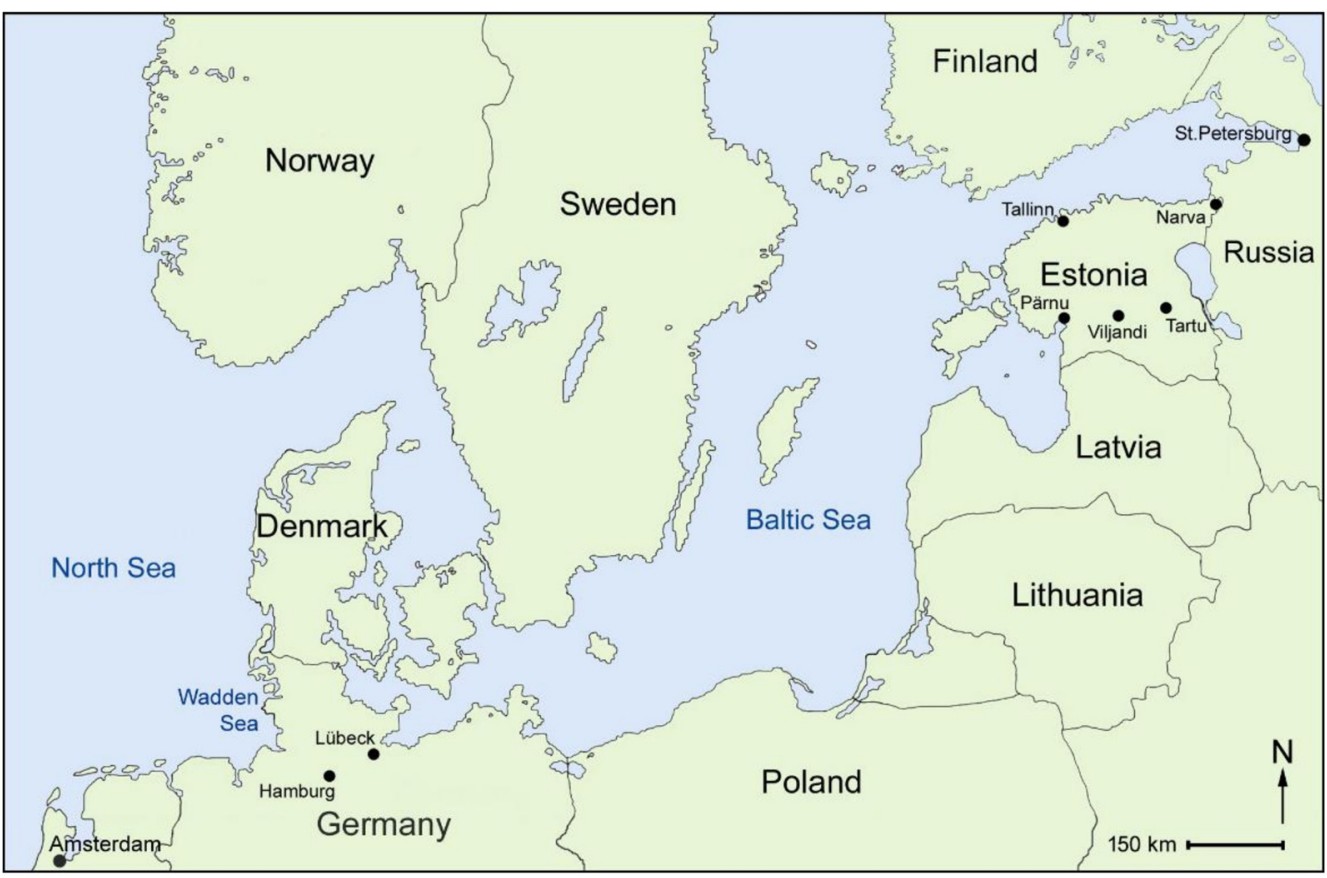

**Figure 1.** Map of the Baltic Sea and locations mentioned in the text. Modified by Lembi Lõugas.

The aim of this article is to combine archaeological and historical evidence on oyster import from the eastern Atlantic, primarily the North Sea, to the eastern Baltic, as well as oyster marketing in late medieval and early modern times, that is, from the 15th to the 19th century. The focus is on oyster shell finds in Estonia, the dietary importance of oysters and whether they were being consumed by the nobility and/or ordinary people. Where our evidence allows, we discuss the origin of the oysters, how they were transported over the Baltic and how they were prepared for the long-distance journey. In this paper, we present the first approach to oyster shell assemblages found in Estonian archaeological collections and combine this information with written sources. The idea is also to document the earliest archaeological evidence and historical record of oysters in the eastern Baltic, particularly in Estonia.

## 2. Archaeological Evidence of Oysters in Estonia

Since the consumption of oysters in the eastern Baltic is quite clearly a rather late (late and post-medieval) phenomenon, the occurrence of archaeological find assemblages providing relevant data correlates with the development of historical archaeology in this region [17–19]. Thus, it is perhaps no wonder that the first noteworthy collections of oyster shells are from the late 1980s and, especially, from the early 1990s, namely from the period when rescue and salvage archaeology on historical sites increased significantly. Whereas oyster shells appear rather sporadically in the urban archaeological collections of the 1990s, this is quite different in the case of the more recently recovered find assemblages: since the early 21st century, but even more actively over the last 10 years, archaeologists have focused their attention on deposits younger than 1600 AD. Nowadays, there is quite a

good basis for research on zooarchaeological material from later historical periods: the Estonian public archaeological collections include dozens of assemblages deriving from urban sites, most notably from the larger towns, for example Tallinn and Tartu, but also from smaller towns, such as Narva, Pärnu, Viljandi, Rakvere and Haapsalu. Less post-medieval artefactual and ecofactual data have been collected in the case of rural areas—here, the previous work has mostly focused on burial sites and/or medieval settlements, but the number of archaeologically investigated manors from the 17th century and later is steadily growing. Accordingly, it is no wonder that the lion's share of our data comes from towns, while potential places of oyster consumption outside the urban settlements (e.g., manors, parsonages, postal stations, taverns, etc.) are under-represented in the present dataset.

### 2.1. Methods of Collection and Research

Oyster shells (or valves) are most often collected manually from archaeological deposits, and no sieves are needed because of the shells' rather large dimensions. Despite criticism of the manual collection of animal remains and recommendations to use fine sieving during the excavations, the need for sieving is minimal for oyster valves selected for sale. However, sieving is a very necessary method for collecting smaller shells, be they oysters or other molluscs [1]. After collecting, the shells are treated and deposited the same way as various other fragile artefacts, preventing the damage that occurs if they are washed by brushing and stored among large and heavy animal bones. In the eastern Baltic, where oysters were definitely being imported, they occur in quite modest amounts on archaeological sites compared with sites of the Atlantic coast; accordingly, it is possible to preserve all of them in the collection repository. As oyster valves have quite a specific shape and differ a great deal from other malacofauna, archaeologists recognize them easily. Archaeological recovery practices have varied: some archaeologists handle shells together with artefacts, while others place them together with ecofacts and store them in the archaeozoological collection.

For the analysis of oyster shells in this study, the online manual compiled by Jessica M. Winder [20], and the methodology described in Campbell [21], Winder [11], Thomas et al. [22] were followed. The analysis included determination of the valve side (right or left), valve shape (round, elongate, irregular or broad), size (height and length in mm), the context and feature, i.e., from which archaeological site and stratigraphic unit the shell comes, while traces of the activity of infesting or encrusting organisms (epibionts) were recorded on a presence/absence basis for shells. Other characteristics, such as the attachment of adult or spat oysters and man-made notches or cuts, were also recorded.

Determining the valve side is important for ascertaining the minimum number of individuals (MNI). In ecological terms, the description of left valves is more important than that of right valves: the left valve is informative, since it rests on the seabed and better reflects the nature of the environment in which it has grown [21,22]. The right side is important when interpreting the consumption of oysters by humans. Oyster shape and traces of epibiont activity are influenced by the environment and reflect the sea bottom depth and conditions in which the oyster lived [21,23]. It is therefore possible to evaluate the origin of shells: whether they are from the deeper offshore zone, from the intertidal zone of the seashore, or from estuaries or creeks. The size of the oyster shell indicates the age (the right valve showing the growth zones better) and growth rate, characteristics that also depend on the environment. If the edge of the shell is broken by less than a third, measurements can still sometimes be taken, as long as the position of the edge can be projected. Analysis of the last annual growth zone provides information about the season of capture [24], but this needs more specific laboratory work and is currently excluded from our study. Moreover, in most cases the outermost layer is broken and difficult to observe.

Find context is important for placing shell finds in the chronological frame and relating them to contemporaneous finds on the same site. In addition, dating on the basis of context makes inter-site comparison easier. Prising notches (opening traces), in the form of V- or W-shaped notches on the shell's edge, indicate where the shell was opened with a knife to

remove the oyster [11,20]. Other opening methods may cause more damage to the edge of the shell, and these kinds of damage due to opening are often impossible to distinguish from ordinary breakage. Notches and cut marks in the interior side show that the oysters were opened while alive [16].

Winder [11], Campbell [21] and Thomas et al. [22] point out that oyster valve size, and shape information, can be used to track the changing management of oyster beds and aid in the reconstruction of trade networks. These authors have developed models for tracing the exploitation of oyster beds: whether the oysters come from natural or cultivated populations, and whether they come from deeper or shallower waters. For such conclusions, we would need quite a large sample size from both natural and cultivated oyster beds for comparison. Since we do not have such material, we cannot develop a detailed discussion of the origin of the oysters solely on the basis of archaeological shell finds.

### 2.2. Material

Our study material consisted of 993 complete, or approximately complete, oyster shells and ca. 302 shell fragments. The largest collections come from archaeological sites in Tallinn, Pärnu and Tartu, while other locations, such as Narva and Viljandi, yielded very few specimens. Of the largest three population centres, Tallinn and Pärnu are situated on the coast and have had adjacent harbours since their inception; Tartu is situated inland but has been a very important trading centre from the beginning of the medieval period, with a connection to Pärnu harbour via the local waterways as well as with Narva via Lake Peipsi [25] (map 36; viabundus.eu, accessed on 1 March 2022). Oyster finds from other places indicate that they were transported to a number of destinations, not only to the large centres.

The analysed oyster shells come from museum collections as well as from recently excavated archaeological material. In Tallinn, the largest collection is stored in the Archaeological Research Collection of Tallinn University (marked by the acronym AI); the collections in Tartu and Pärnu are held in the local museums (TM and PäMu, respectively). Smaller collections are also available in other repositories. All obtained data and photos of the shells are available in the repository of research data [14] (see Supplementary Materials).

### 2.3. Results of the Analysis of Oyster Shells

Summarized results of the analysis of oyster shells found in archaeological deposits in Estonia are presented in Table 1. The locations of archaeological sites are grouped by towns, where the number of excavated sites indicates how many of them were included in this study. The date, according to the context, is also given, divided into three time periods, based on the historical development of the region: from the 15th to mid-16th century, representing the medieval "Hanseatic" period; the mid-16th to the 17th century (ca. 1550–1700), the time of Swedish rule; and the 18th to the 19th century, the time after Estonia was incorporated into the Russian Empire. It must be mentioned that no radiocarbon dating was performed on the shell (from the mineral compound or preserved ligaments), and that all the dates are estimates based on archaeological context information, i.e., according to accompanying find types (mostly fragments of pottery and clay tobacco pipes). Further, the numbers of more or less complete left and right shells and the numbers of shell fragments are shown. The larger number in the NISP 1 column (Table 1) also indicates the minimum number of individuals (MNI).

The range of measurements gives some idea of shell sizes, but as ecological assessment of the shells' origin is not included in the current study, statistical evaluation has so far not been undertaken.

**Table 1.** Oyster valves and fragments from archaeological deposits in Estonia, sorted by context.

| Location | Date by the Context | NISP 1 S/D | NISP 2 | Valve H (mm) min–max | Valvel L (mm) min–max |
|---|---|---|---|---|---|
| Tallinn (45 sites) | 15th–16th century | 3/2 | 29 | 46–111 | 64–103 |
| | 16th–17th century | 25/28 | 24 | 44–112 | 39–98 |
| | 18th–19th century | 44/54 | 31 | 35–107 | 31–101 |
| | N/A (14th–19th c.) | 24/36 | 63 | 41–108 | 33–95 |
| Tartu (19 sites) | 16th–17th century | 5/6 | 5 | 62–85 | 55–68 |
| | 18th–19th century | 79/88 | 15 | 47–121 | 44–100 |
| Pärnu (7 sites) | 15th–16th century | 0/1 | 4 | 78 | 80 |
| | 16th–17th century | 85/84 | 38 | 50–106 | 43–100 |
| | 18th–19th century | 175/254 | 93 | 41–117 | 34–109 |
| Other | 17th–19th century | ca. 30 | | N/A | N/A |

NISP 1 = number of identified (complete, or approximately complete) specimens (S – left shell / D – right shell); NISP 2 = number of identified (broken) specimens; Valve H = height in mm; Valve L = length in mm.

### 2.4. Variation in Oyster Shell Characteristics

According to Winder [20], Campbell [21] and Thomas et al. [22], an oyster's shape is influenced by the environment in which it grows. For example, a round shape forms in slow tidal currents, an elongate shape forms in deeper, offshore environments, while an irregular shape forms on a rough or uneven surface. Greater variation in oyster shells may indicate the exploitation of different oyster populations, while a more homogeneous size and shape may point to cultivated populations. In general, our study results show variations in shell shapes, though on some sites the variation is not very notable. This may indicate shells originating from oysters that have been selected before consumption. Most of the oyster shells in our collection are of regular shape (slightly oval) or round (Figure 2a,b). Only a few elongate and broad (Figure 2c) variants have been recorded, as well as a few old individuals (Figure 2d). Additionally, quite a few irregular-shaped shells were found, while many regular or round shells had irregular heels (Figure 2c).

Greater variability was observed in the traces left on the shells by epibionts. Winder [11,20] describes many types of evidence found in the shells. She mentions that the most common are marine polychaete worms (*Polydora* sp.), which burrow into the general outer surface of the shell (Figure 3a). Additionally, a sponge, *Cliona celata*, leaves round holes that perforate the shell (Figure 3b). Some marine worms leave traces in the form of calcareous or sand tubes in the outer surface of the shell. Readily identifiable are barnacles (*Balanus* type), which in many cases remain attached to the oyster shell but can also be represented merely by traces on the shell (Figure 3c,d). Bryozoa are minute invertebrates occupying individual box-like cells that are joined together in large colonies, appearing to the eye like moss or lace on the shell (Figure 3e). On the basis of epibiont traces, it is possible to distinguish the origin of the shell: whether it is from an intertidal, littoral or shallow sublittoral bed. In our collection epibiont activity has been recorded in 188 cases (69 shells from Tallinn, 59 shells from Pärnu and 60 shells from Tartu have such traces). However, we cannot draw any firm conclusions on the precise conditions in which they grew, though we can conclude that they come from shallower rather than deeper waters.

In addition, other qualitative characteristics, such as the attachment of adult or spat oysters, and man-made notches and cuts, have been recorded in our dataset [14] (see Supplementary Materials). Valves with additional oysters growing on the shell (Figure 4a,b) are indicative of a habitat in a natural bed, where cultivation was absent or limited [11,21]. It is unclear whether the conclusion can be extended to all of the oyster remains found in Estonia, but at least some shells with additional attached shells have been recorded, indicating exploitation of natural rather than cultivated beds.

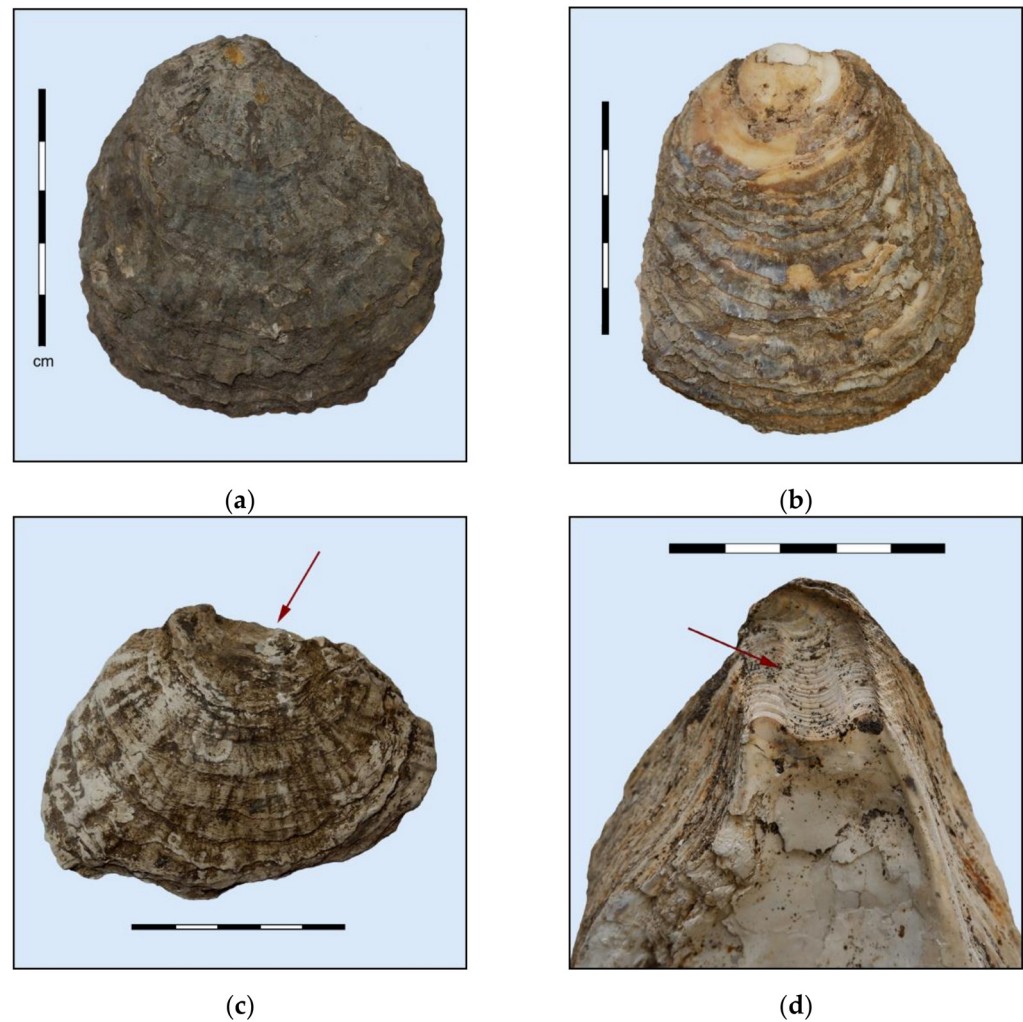

**Figure 2.** Oyster shell halves from archaeological sites in Estonia: (**a**) left shell of regular shape, external view (PäMu 40372 A 2698:500/19), (**b**) right shell of regular shape, external view (AI 6965:105), (**c**) left shell, broad with irregular heel, external view (PäMu 40372 A 2698:500/63), (**d**) right shell of an old specimen, internal view (AI 7575:2311). Photos: Lembi Lõugas.

The edges of most of the shell halves in our study have been broken to a greater or lesser extent, probably not only because of opening the oyster but also due to post-excavational handling. Fewer have breakage in the form of V- or W-shaped notches, i.e., opening marks on the oyster shell made with a knife. Most opening marks on the archaeological shells are on the ventral edge, farthest from the hinge, while modern-day opening uses a short blade from the posterior edge, which is the shortest distance to sever the adductor muscle. In many cases in this study, it was impossible to interpret the edge damage as being due to opening rather than just ordinary breakage. However, this may also be the result of a different opening method. At all Estonian archaeological sites, clear notches and cut marks were found on at least a few shells (Figure 5a,b), which shows that at least some oysters were opened while alive and possibly eaten raw. Many shells have a brown iridescence on the internal surface while other shells were blackened, which is thought to be caused by burning, both being possible signs of cooking in the shell (Figure 5c). Such colouring is noticed especially in the 15 Lai Street collection from Pärnu [26]. If oysters are to be cooked, then roasting them in ashes or ovens in moderate heat causes the valves to open automatically as the animal is poached in its own liquor [16].

The opening of oyster shells with a special knife, or heating them on a fire, are not the only known methods of getting access to the flesh. Cracking the oysters by striking the ventral end with a round cobble or flat stone [27,28] is a somewhat more archaic method, but the traces of some kind of impact at the ventral end are quite similar to those found on many specimens in our collection (Figure 5d). Thus, according to shell finds from Estonian archaeological sites, the most common methods of opening oyster shells seem to be heating and cracking.

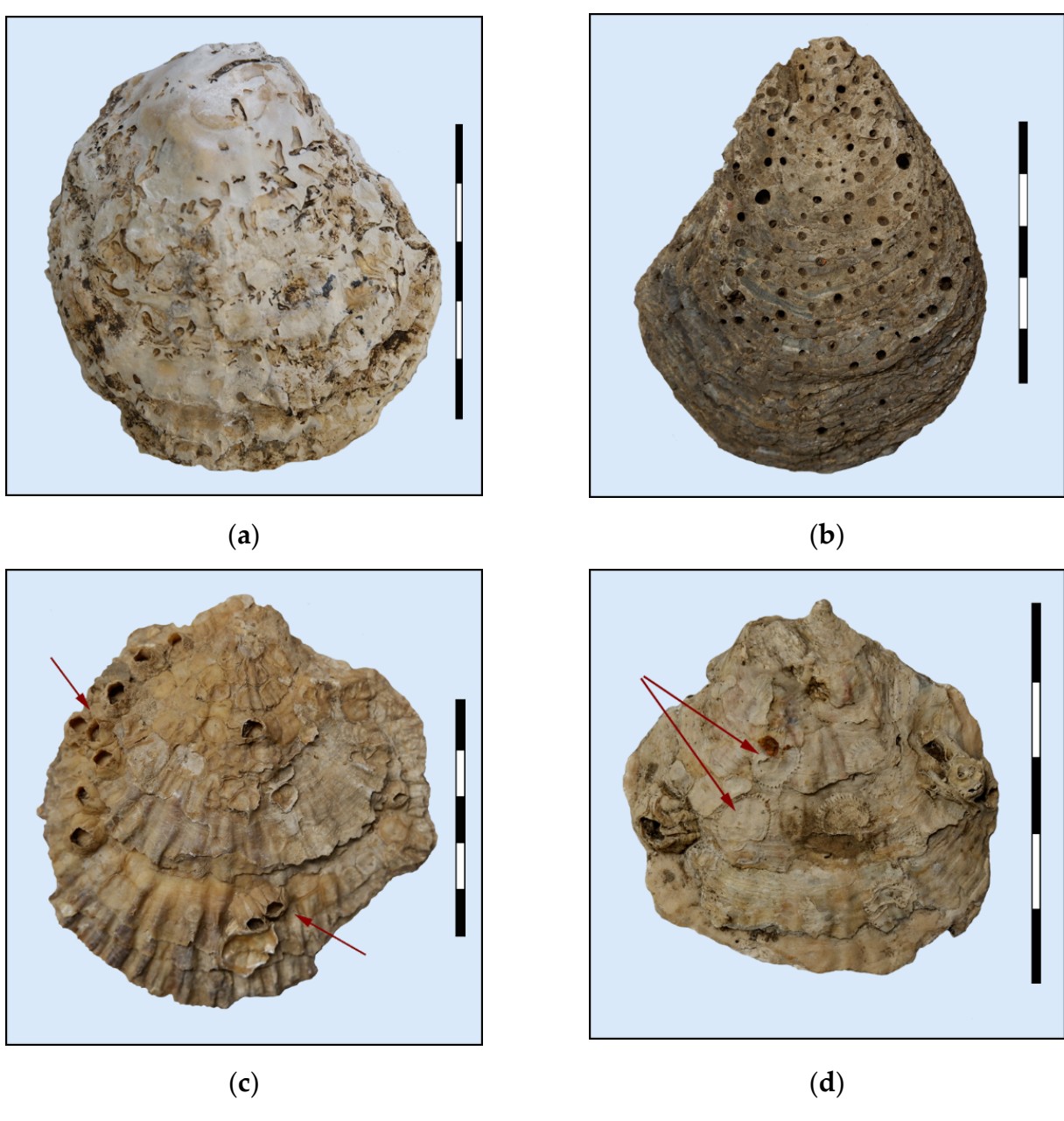

(**a**)  (**b**)

(**c**)  (**d**)

**Figure 3.** *Cont.*

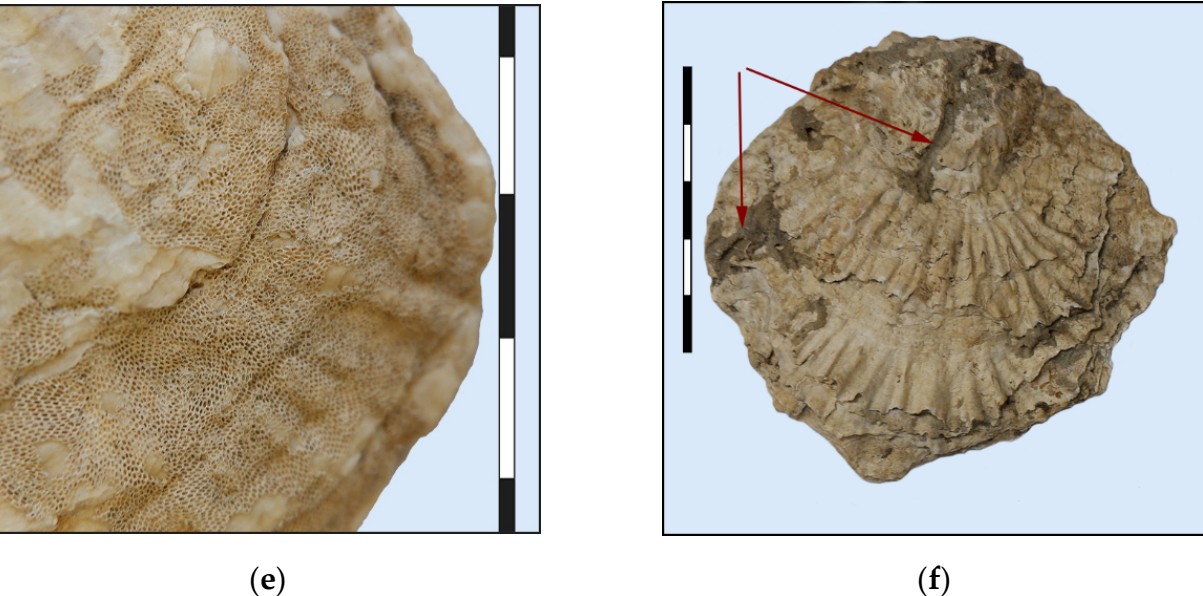

(**e**)  (**f**)

**Figure 3.** Traces of infesting and encrusting organisms (epibionts) on oyster shells found in Estonia (identified according to Winder [18]): (**a**) burrows of marine polychaete worms *Polydora* sp. (AI 7575:3511), (**b**) borings of the sponge *Cliona celata*, showing a honeycomb-like appearance (TM A 73:8), (**c**) barnacles, *Balanus*, in many cases attached to the shells (TM A 116:2079/2), (**d**) traces of barnacles on the shell (TM A 116:1702), (**e**) "pattern" left by Bryozoa on the shell (AI 6787:4/1), (**f**) sand tubes of Sabellid worms (PäMu 14350 A 2501:46/10). Photos: Lembi Lõugas.

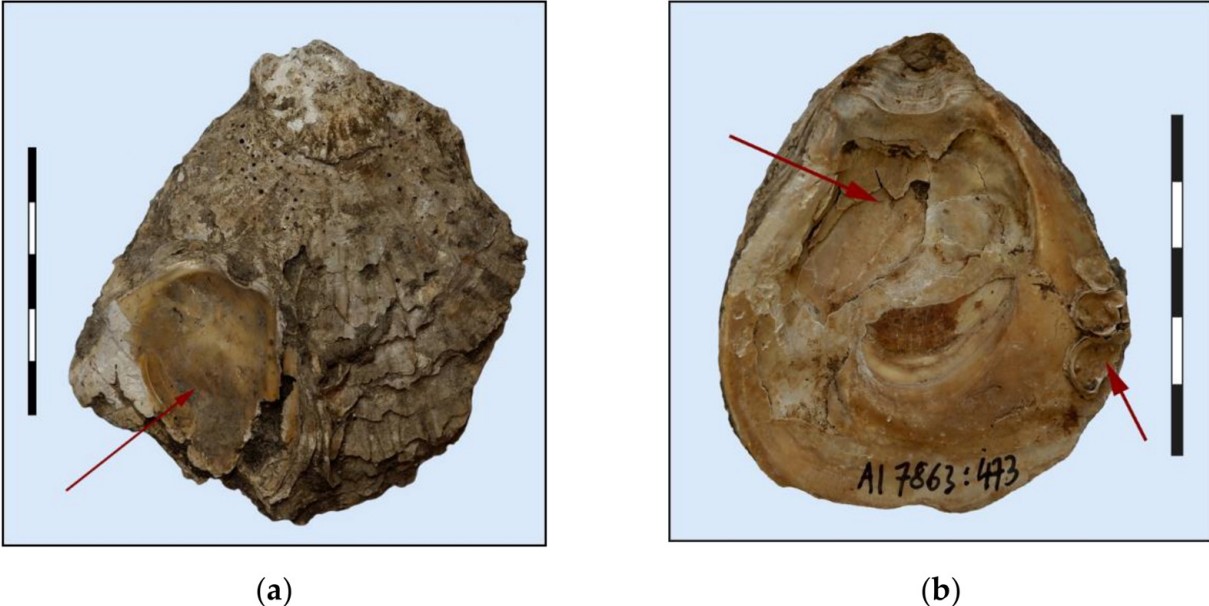

(**a**)  (**b**)

**Figure 4.** Traces left by natural processes found on the archaeological specimens in Estonia (identified according to Winder [18]): (**a**) oyster shell attached to the external surface of another shell (PäMu 40372 A 2698:500/57), (**b**) young oysters attached to the internal surface of the right valve and chambering formed during rapid salinity changes (AI 7863:473). Photos: Lembi Lõugas.

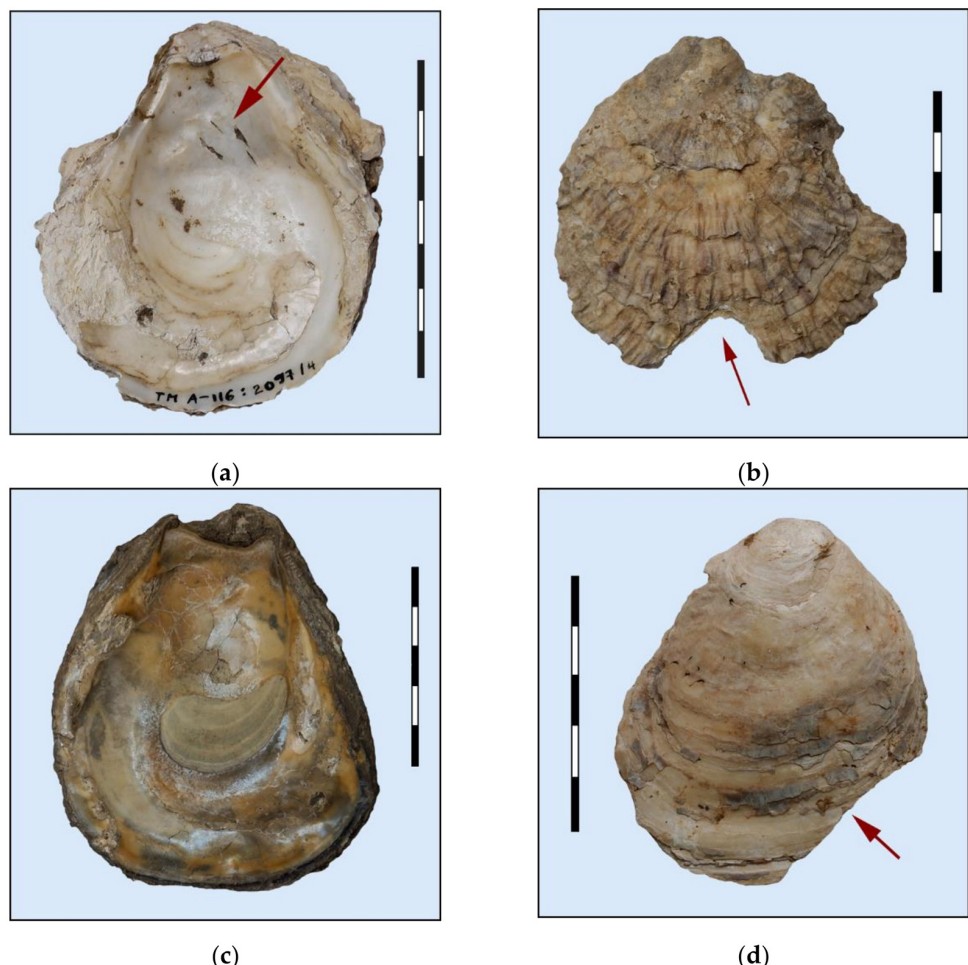

**Figure 5.** Man-made marks, notches and cuts found on the archaeological specimens in Estonia (identified according to Winder [18] and Rosser & Rick [26]): (**a**) cut marks on the internal surface of the shell (TM A 116:2097/4), (**b**) notches made when opening the oyster (TM A 73:2), (**c**) brown-black colour indicating roasting of the oyster at moderate intensity (PäMu 40372 A 2698:500/41), (**d**) straight breakage formed when opening the oyster by cracking (AI 8013:3282). Photos: Lembi Lõugas.

## 3. Historical Evidence of Oysters in Estonia

From the second half of the 17th century onwards, oysters appear regularly in various archival sources [29] (p. 16); [30] (p. 68). Quite certainly, the first oysters arrived in Estonia before this, but initially their import was scanty and random, and they had no impact on local gastronomy. Considering the fact that oysters were not stored, even for the most grandiose feasts in medieval Tallinn, which was the biggest and wealthiest Hanseatic town of Estonia, it is highly likely that oysters were not being imported to Estonia at that time. The long period of the Livonian War in the second half of the 16th century, and further conflicts at the beginning of the 17th century, brought impoverishment and famine; hence, the times were unsuitable for enjoying culinary pleasures and luxurious foods. Only after the stabilization of the political and economic situation could people pay attention to new developments in gastronomy and enlarge the assortment of imported foods.

Various account books preserved in Estonian archives provide information about the import and consumption of shellfish in the early modern period. Overseas trade in the 17th and 18th centuries mainly took place through the harbours of Tallinn, Narva and Pärnu. Therefore, customs books contain essential data on shellfish import (for example, on the amount of import, prices of goods and so forth) but they also reveal the arrival time and departure port of seagoing ships. Diverse feast accounts from the Tallinn City Archives provide more specific information about the consumption of oysters.

Nevertheless, oyster consumption always remained limited in Estonia. Imported shellfish were valued only by wealthy citizens and the nobility. Oysters were consumed as a delicacy and luxury food on special occasions only. However, according to written sources the local elite consumed shellfish in quite remarkable quantities. Oysters appear in account and customs books throughout the year, including the summer period. For example, on 8 July 1669, at a feast given in Tallinn in honour of state counsellor Lars Fleming, three big and four small barrels of oysters were also served, among other delicacies [31] (p. 255). In the 17th and 18th centuries, oysters were obligatory at the feast of St Thomas, given by the Tallinn Town Council [31] (p. 44, 46, 186, etc.). For that feast 100–200 oysters were usually bought, which were most probably served with lemon [31] (p. 48, 48v).

Already by the second half of the 17th century, eating oysters had become quite popular among the social elite in Estonia, and demand for shellfish increased. Oysters were mostly imported in small barrels and casks, but bigger vessels were sometimes also used. For example, in 1781, two and a half barrels contained as many as 3000 oysters [32] (p. 24). It should be mentioned that accurate records were kept when selling and buying. In 1771, no less than 126836 fresh or raw oysters were imported to Tallinn [33] (p. 412). In October 1682, Tallinn Town Council alone bought oysters seven times, in 10 small barrels altogether [31] (pp. 4–5). Afterwards, the consumption of oysters spread to manors and small towns as well. Accounts and correspondence preserved from the second half of the 18th century illustrate how two big trading companies in Pärnu, Jacob Jacke and Company and Hans Diedrich Schmidt, supplied numerous manors in the surrounding counties, and some small towns, with shellfish [34]. According to written sources, these were often fresh or raw oysters (*100 st frische östers*) [31] (p. 6v), which means that transporting them from the harvesting place to Estonia must have been very well organized, for otherwise the food could easily have become spoiled. It is known that oysters remain fresh for up to 10 days, or up to 8–12 weeks, if kept cool and closely packed [35]. They could have simply been tightly packed in baskets or barrels to keep the valves tightly shut, preventing desiccation [10]. This would have been enough time for the voyage across the Baltic Sea and sale in markets.

Oysters reached the consumers in Estonia not directly from their harvesting place but often through various intermediaries. Seagoing ships loaded with shellfish mostly arrived from Lübeck and Amsterdam. According to the customs books of Pärnu, in addition to merchants, ordinary people imported oysters as well, but only for personal use [36] (p. 46). Usually, there is no information preserved in archival sources about the origin of oysters imported to Estonia, with the exception that, in October 1781, the customs book of Pärnu records that 400 Holstein oysters were imported from Lübeck [32] (p. 24). According to information from the 19th century, in Schleswig-Holstein there were oyster beds near Amrum, Föhr, Hallig and Sylt, in north-western Germany [37] (p. 68).

18th century customs books reveal that, in addition to fresh or raw (live) oysters, salted oysters (*gesaltzene Austern*) and conserved (perhaps marinated) oysters (*eingemachte Austern*) were being imported to Estonia [38] (pp. 4, 35, 36). The conserved oysters were transported in small glass vessels (jars) (*10 Gläser eingemachte Austern*), which indicates that they were preserved without shells [39] (p. 7). At the same time, salted oysters were sold in barrels; hence, they were conserved with shells or at least half shells. The 18th century Baltic German commentator, August Wilhelm Hupel, wrote that oysters were popular with the German elite in Estonia, bringing a price of 4 roubles for 100 pieces [33] (p. 461). Conserved shellfish (oysters and mussels) were also highly valued as imported goods. According to Hupel, foreigners also tried to cook local Estonian molluscs for meals, and regarded some of them as quite tasty. Nonetheless, local people were not ready to value them as food.

## 4. Discussion

Most of the archaeological oyster shells in Estonia come from archaeological assemblages dating approximately from the second half of the 17th, and up to the 18th, century. The oldest finds may be from the late 15th century; however, the find contexts are usually

too vague to be absolutely certain. It is interesting to note that quite often the shells are accompanied by fragments of clay tobacco pipes—whether this is a marker indicating the site of a tavern/inn, or just evidence of feasting, needs further analysis in the future.

The broad analysis of the sites where the oyster shells have been recovered offers, by and large, just a few surprises. As expected, most of the data comes from the major urban centre of Estonia, Tallinn, which has been archaeologically investigated much more intensively than other places. Our database consists of 45 individual investigations that can be topographically divided into three main areas: three sites are situated in the upper town (*Toompea*, in Estonian), which was the administrative centre of the province, both during the period of rule by the Kingdom of Sweden (1561–1710) and under the Russian Empire (1710–1917). The upper town can also be regarded as the residential area of the nobility; although, in addition to the elite, people of lower social status (such as servants), as well as various craftsmen, also lived here. Next, there are nine sites in the heart of Tallinn, within the walled Hanseatic town, which was inhabited by very diverse strata of townsfolk. Last but not least, most of the oyster finds come from the various suburbs of the city, where 33 fieldwork projects have produced relevant material.

In general, the above-mentioned statistics are not surprising. First of all, research in the upper and lower town has been less extensive than in the suburbs, as the historical core of Tallinn is relatively well preserved. In addition to this, one has also to bear in mind that much of the urban waste of the town plots was not accumulated on the spot but was removed from the fortified centre of the town. Urban waste management is visible in the archaeological record in at least two ways: the collected muck was distributed on the fields and gardens of the suburban plots around the southern side of the town as a fertilizer (among other examples, perhaps most vividly described in [40,41]), or else the refuse deposits were taken to a designated area north of the walled town and used in the reshaping of the suburban townscape [42]. Thus, the exact dating of the assemblages is, in most cases, difficult. The *intra muros* finds are mainly from mixed layers, not from sealed contexts; the suburban material, on the other hand, is either thoroughly ploughed and/or comes from tertiary layers (e.g., refuse from unidentified town plots). Therefore, providing precise dates for the consumption of the oysters generally involves a strong element of subjectivity.

However, assessment of the associated artefacts from the same layers does offer at least some basis for interpretation. In the upper town, the earliest find complexes with oyster shells appear to date from the 17th century (e.g., material from 17 and 21/2 Toom-Kooli Street). The situation appears to be similar in the lower town, where, along with undatable contexts (heavily mixed layers or stray finds), there is evidence from two sites (9 Kooli Street and 52 Lai Street/5 Tolli Street) to suggest that the food waste is from the second half of the 17th century, and in one case (7 Aia Street/12 Uus Street), perhaps even from the first half of the 17th century. In the suburbs, the majority of this evidence can be associated with the 18th and 19th century, and in some cases, the late 17th century (e.g., Roosikrantsi Street, 7 Estonia Boulevard and 17 Põhja Boulevard/1 Soo Street) is not out of the question. Whether we are dealing here with suburban plots of higher echelons of the townsfolk (many had summer homes outside the city walls), or whether this should be interpreted as rubbish from the town centre, needs further analysis in the future.

There is one suburban area that needs extra attention: the Kalamaja suburbs, especially around Jahu Street in the north-western part of Tallinn. Two sites in this district, north of the town wall, have offered remarkably early dates for the consumption of oysters when compared with the overall data. At 5 Jahu Street, from the activity layers associated with a house of the early modern period, there are some oyster shells from the 17–18th century, and in one case, even the 16th century is a possibility. Still, since we do not have sealed and well-dated contexts here, we must be cautious about dating these finds to such an early period with absolute certainty. Then again, such an early date is perhaps not absolutely impossible, if we consider the collection unearthed across the street, at 6 Jahu Street. Here, a late medieval landfill with the peak of deposition around the 1480s [42,43]

also yielded 43 oyster shells. Whereas most of the shells can be associated with later times, the 17–18th century, there are a couple of finds that might come from late 15th and early 16th century refuse deposits. At least based on the layer descriptions and associated artefacts, a later dating seems to be ruled out, yet the nature of the site (landfill) and the research method (salvage excavation) leaves plenty of room for speculation. Nevertheless, since these deposits also include other kinds of late medieval luxury food remains (such as walnuts) and items of higher social status, it is not completely out of the question that we may be dealing here with the earliest evidence of oyster consumption in Tallinn, if not in the eastern Baltic as a whole.

Moving now to Tartu, we find that the evidence from the second-largest town in Estonia does not add much to the variety of consumption patterns or relative chronology of the appearance of oysters in the archaeological record. The catalogued finds are comparatively evenly distributed across the townscape, with 19 sites in the town core and nine outside the former defence line, i.e., in the suburbs. The inner town collections are mostly from traditional consumer contexts (e.g., town plots) without any significant bias towards the highest strata of the townsfolk. Generally, the oyster finds are associated with artefacts of the late 17th to 18th century and later, and none of the registered find contexts seem to belong to earlier settlement phases. In one noteworthy exception, two oyster shells were found in a later 18th century burial vault inside St John's Church, perhaps indicating disturbance of the chamber in the 19th–20th century.

The third largest collection of relevant finds comes from Pärnu, more precisely from the area of the former Hanseatic town, New Pärnu. Currently we have data on seven sites. The overall number of excavations with oyster finds is considerably broader but does not change the general impression obtained in the course of research for the present paper. In contrast to Tallinn and Tartu, all finds are from the town core, inside the medieval town walls, and not from historical suburbs. Chronologically, the majority of the collected oyster shells appear to have been discarded in the 18th century, and larger-scale consumption seems to begin in the late 17th century [14]. Although none of the sites have revealed finds earlier than 1600 AD, there are general indications of oyster consumption in the late 16th century: in some cases, the assemblages were dominated by fragments of 16th century pottery and stove tiles, though it should be noted that these had been redeposited during the 17th century reconstruction of the townscape.

There is one find context that deserves separate attention: one of the most exceptional oyster assemblages studied thus far comes from 2 Munga Street in Pärnu. Here, inside a post-medieval log building, a wooden tub with a diameter of 60 cm was unearthed, which had been sunk into the floor of the log house [44] (p. 13). More than 200 unopened oysters were collected from this tub, along with the lower part of a wine bottle. Based on the bottle, the collection of oysters should be dated to the latter part of the 18th century or the first decades of the 19th century. The director of the rescue excavation has not offered any interpretation of this feature, but it seems highly likely that we could have here the only surviving example of pre-consuming deposition at a consumer site. This oyster collection merits more detailed research and the use of precise methods like the morphometric study (see e.g. [21]), stable isotope analyses to establish sea temperature, season of harvest and provenance in Europe.

To sum up the archaeological data on oyster finds, it appears, based on the surveyed collections, that the peak of oyster consumption was during the 18th century, this culinary habit having started to develop on a larger scale in the 17th century, and, stated with some caution, the very first evidence of oysters might belong to the late medieval period, i.e., the latter half of the 15th century.

## 5. Conclusions

In accordance with the objectives of this article, which were to combine archaeological and historical evidence on oyster import from the North Sea to the eastern Baltic and oyster marketing in medieval and early modern Estonia, we can conclude that the oysters sold

and consumed here were most probably harvested in the Wadden Sea. The shellfish were provided for consumption not by ordinary people, but by the nobility. One of the issues at hand was to discuss long-distance transport and whether fresh oysters would have survived such a long-lasting journey. The archaeological and written evidence shows that this did occur, but at the same time, prepared oysters were also valued. Many archaeological oyster shells have marks of heating or roasting, which may simply indicate the method of opening the shell halves, or that fresh oysters were being kept too long before consumption, since roasting is a good way to prolong the shelf life of this food. The trade in oysters to Estonia was most intensive in the 17th and 18th century, as indicated by the archaeological and written sources, but a few archaeological shell finds also come from contexts dated to the end of the 15th and the 16th century. In addition, from Pärnu, there is a find of a wooden tub filled with unopened oysters and a wine bottle, dated to the 18th century. This exceptional find allows a more complex case study of oysters as one of the luxuries of the eastern Baltic.

**Supplementary Materials:** The following supporting information can be downloaded at: https://datadoi.ee/handle/33/442 (accessed on 11 March 2022).

**Author Contributions:** Conceptualization and methodology, L.L., I.J., E.R.; investigation, L.L., I.J., E.R.; data curation, L.L.; writing—original draft preparation, L.L., I.J., E.R.; writing—review and editing, L.L., I.J., E.R.; visualization, L.L.; project administration, L.L. All authors have read and agreed to the published version of the manuscript.

**Funding:** This research was funded by the Estonian Research Council, grant number PRG29.

**Institutional Review Board Statement:** Not applicable.

**Informed Consent Statement:** Not applicable.

**Data Availability Statement:** Data generated during this study is available in the Datadoi repository of Tartu University [14].

**Acknowledgments:** We thank Arvi Haak for his kind permission to use the archaeological collections of Tartu City Museum, Margo Samarokov for use of the collections in Pärnu Museum and Valdis Bērziņš for English proofreading.

**Conflicts of Interest:** The authors declare no conflict of interest.

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
