# Peer review of "European Flat Oyster (Ostrea Edulis L.) in the Eastern Baltic as Evidence of Long-Distance Trade in Medieval and Early Modern Times"

_heritage, doi:10.3390/heritage5020044_

Round 1

Reviewer 1 Report

This is a very thoroughly carried out contribution which combines archaeological and historical records in an interesting and useful way. Although there are limitations on the zooarchaeological dataset, most notably the lack of direct dating, these are acknowledged (and to be expected given the nature of the original excavations).

I do not have any substantive comments to make about the paper, although I have a small number of corrections as follows:

Line 15 ‘mollusc’ not ‘mussel’

Line 31 ‘mollusc’ not ‘mussel’

Line 37-38 I don’t agree that mussel is an appropriate term here. Mussels are specifically mytiliform molluscs. Stick to shellfish, or bivalves perhaps?I won’t make more reference to this but my feeling is that all references to mussels in this paper should be changed.

Line 127 delete ‘the’ in front of ‘deposits’

Line 135 delete ‘the’ in front of ‘towns’

Author Response

The authors thank the reviewer for valuable comments on our manuscript. All comments have been accepted and the text has been corrected accordingly:

Line 15 ‘mollusc’ not ‘mussel’ - corrected

Line 31 ‘mollusc’ not ‘mussel’ - corrected

Line 37-38 I don’t agree that mussel is an appropriate term here. Mussels are specifically mytiliform molluscs. Stick to shellfish, or bivalves perhaps?I won’t make more reference to this but my feeling is that all references to mussels in this paper should be changed. - corrected

Line 127 delete ‘the’ in front of ‘deposits’ - corrected

Line 135 delete ‘the’ in front of ‘towns’ - corrected

Reviewer 2 Report

A good and interesting paper already, that could be great with a few changes (the two critical points are noted in the line-by-line comments below). A slight distraction is that some of the sentences are rather convoluted in a Germanic academic style; most English readers expect these thoughts as consecutive shorter blunter sentences.

Line-by-line comments: Lines in the document are specified here by numbers in [square brackets]. The critical points are those for lines [70]-[72] and [144].

[2] and elsewhere: check through the text to make sure there is NO comma between edulis and L. or Linnaeus. The comma is only allowed between the cited authority (here, Linaeus) and the date of the work of that authority if it is included (here, 1758), but confusingly the date reference to that published work is not required, and is never cited in full (in biological literature the reader is expected to already know the published references for citing authorities: every biologist is supposed to somehow know already that the 1758 in Ostrea edulis Linnaeus, 1758 means the 10th edition of his Systemae Naturae published in Stockholm).

[37] and elsewhere: in English, the word ‘mussel’ does not mean shellfish or shells in general, it means only shellfish in the Family Mytilidae (the genera Mytilus, Modiolus, Musculus, Brachidontes, Perna), usually with black or purple shells. The use of the word ‘muschel’ to mean shells in general is solely in those German dialects descended from Old High German (OHG ‘muscula’ from the Latin musculus, which also meant Mytilidae),. Most Romance languages use a descendant of the Latin term concha which Romans used for shells in general (Spanish/ Portuguese ‘concha’, Italian ‘conchiglia’), while ohers use descendants of cochlea, the Latin for shell/coiled thing/ear, derived from the Greek κοχυλί (Occitan ‘cauquilha’, langue d’oil ‘coquille’). North Atlantic European languages including English do not use these words, they use words derived from Old Norse skal, this includes ‘shell’, the word used in English everywhere in the World. Please check through the text to use the word ‘shell’ unless you really mean a Mytilidae ‘mussel’.

[56] ‘cowries most abundant in Indian Ocean’: the Atlantic and Mediterranean have lots of cowries and lots of different kinds, the Mediterranean forms are quite big and shiny. What is interesting is that these cowries closer to the Baltic are avoided, in favour of the Indian Ocean cowries. Please make it clear that the cowries imported to the Baltic are the Indian Ocean species.

[70] – [72]: ‘As the conditions in the Baltic Sea are not suitable for oysters...’ This assertion is critical for the entire argument in this paper. It therefore must be established beyond any reasonable doubt. (It’s true, but the point is not made here robustly enough to convince archaeologists unfamiliar with shells, which are almost all of them.) Since we are dealing with the past, the authors must show the oyster could never have been in the Baltic in its past. So this section must be expanded by a sentence or two, quoting some references in the biological literature to oysters’ tolerances for salinity and sea temperature, and also quoting some references in the oceanographic literature which demonstrate that the Baltic’s salinity and sea temperature get too low for oysters (how often do eastern Baltic ports freeze over?). A quick search on Google Scholar for ‘Ostrea edulis salinity’ and ‘Ostrea edulis temperature’ revealed several useful papers, often Swedish research.

[74]: another ‘mussel’ when the authors mean ‘shell’.

[144] ‘… the need for sieving is minimal for oyster valves [1].’ This both incorrect, and the opposite of the conclusion in Reference [1]. Oysters when they are well-preserved can be as small as a thumbnail, and when they are not well-preserved the pieces of large oysters bearing the hinge (which are identifiable and countable portions of valves, left or right) can be as small as a little fingernail. Not sieving for oysters gives a massive overstatement of their average sizes, and a massive underestimate of their numbers. The lack of sieving also explains why so few other kinds of shells, especially Mytilid mussels, are found by archaeologists (line [75] of this paper). Sieving for shells including oysters is absolutely necessary, and this paper will help put Estonian archaeology ahead of the rest of Europe if it says so.

[184] ‘notches and cut marks on the interior side show that the oysters were opened while alive and eaten raw’. Shucking-marks (in North American English) only show the oyster was opened, not that it was eaten raw (the authors know this already, they explain it is only possible on Line 289).

[189] ‘models for tracing the exploitation of oyster-beds ...’ The main reference for using oyster shape is in fact Reference [21], so a reference to it should be included here. Oysters grade in shape from broad, frilly-edged with small hinges in shallow muddy beds to tall, boat-shaped with big hinges in offshore beds with strong currents. Most of the oysters in this paper have the shape of oysters from shallow beds in moderate currents.

[223] this sentence on the reservoir effect can be deleted. Almost no archaeologists will expect radiocarbon dating on strata this recent, and even fewer will expect radiocarbon dating of the shells directly. If the sentence is included, what this reservoir effect is needs explaining, which will be complicated.

[233]: see the note above on page [189]; reference to Reference [21] needs including here.

[263] the sentence starting ‘However, without knowing the harvesting areas ...’ can be deleted. Nobody knows exactly which beds were harvested for archaeological oysters, the nature of the bed is always all that can be reconstructed. Striking the word ‘only’ in the next sentence ‘We can only conclude ...’ shows what the authors have done, and that they have done it well here.

[285] Most archaeological opening marks in all periods are on the ventral edge, farthest from the hinge, while modern-day opening uses a short blade from the posterior edge (the inward curve) which is the shortest distance to sever the adductor muscle: a note in this paper on how often the stab-wounds are on the rear of the shell in the modern way would help archaeologists know where and when this way of oyster-opening began.

[308] another use of ‘mussel’ when ‘shell’ is meant. Here the word ‘mussel’ can be replaced by ‘flesh’ or ‘meat’.

[308]-[313]: this is the first observation of oyster-’cracking’ as an opening-method in archaeological material known to this referee, who will watch for this in his oysters and refer to this paper when he finds any. A good reason to publish this paper, editor!

[482] – [490] As an aside, the barrel of oysters from 2 Munga Street Parnu is exceptional, internationally important, and merits further research (detailed morphogmetric study as in Reference [21]; isotopic analysis to establish sea-temperature, season of harvest and provenance in Europe).

Author Response

Reviwer 2

The authors are very grateful and thank the reviewer for valuable comments on our manuscript. These allowed us to improve the text and correct some errors. All comments have been accepted and the text has been corrected accordingly:

[2] and elsewhere: check through the text to make sure there is NO comma between edulis and L. or Linnaeus. The comma is only allowed between the cited authority (here, Linaeus) and the date of the work of that authority if it is included (here, 1758), but confusingly the date reference to that published work is not required, and is never cited in full (in biological literature the reader is expected to already know the published references for citing authorities: every biologist is supposed to somehow know already that the 1758 in Ostrea edulis Linnaeus, 1758 means the 10th edition of his Systemae Naturae published in Stockholm). - Corrected
[37] and elsewhere: in English, the word ‘mussel’ does not mean shellfish or shells in general, it means only shellfish in the Family Mytilidae (the genera Mytilus, Modiolus, Musculus, Brachidontes, Perna), usually with black or purple shells. The use of the word ‘muschel’ to mean shells in general is solely in those German dialects descended from Old High German (OHG ‘muscula’ from the Latin musculus, which also meant Mytilidae),. Most Romance languages use a descendant of the Latin term concha which Romans used for shells in general (Spanish/ Portuguese ‘concha’, Italian ‘conchiglia’), while ohers use descendants of cochlea, the Latin for shell/coiled thing/ear, derived from the Greek κοχυλί (Occitan ‘cauquilha’, langue d’oil ‘coquille’). North Atlantic European languages including English do not use these words, they use words derived from Old Norse skal, this includes ‘shell’, the word used in English everywhere in the World. Please check through the text to use the word ‘shell’ unless you really mean a Mytilidae ‘mussel’. - Corrected

[56] ‘cowries most abundant in Indian Ocean’: the Atlantic and Mediterranean have lots of cowries and lots of different kinds, the Mediterranean forms are quite big and shiny. What is interesting is that these cowries closer to the Baltic are avoided, in favour of the Indian Ocean cowries. Please make it clear that the cowries imported to the Baltic are the Indian Ocean species. - Latin name is added to „money cowry“ and the reference indicating the origin of Estonian finds is added.

[70] – [72]: ‘As the conditions in the Baltic Sea are not suitable for oysters...’ This assertion is critical for the entire argument in this paper. It therefore must be established beyond any reasonable doubt. (It’s true, but the point is not made here robustly enough to convince archaeologists unfamiliar with shells, which are almost all of them.) Since we are dealing with the past, the authors must show the oyster could never have been in the Baltic in its past. So this section must be expanded by a sentence or two, quoting some references in the biological literature to oysters’ tolerances for salinity and sea temperature, and also quoting some references in the oceanographic literature which demonstrate that the Baltic’s salinity and sea temperature get too low for oysters (how often do eastern Baltic ports freeze over?). A quick search on Google Scholar for ‘Ostrea edulis salinity’ and ‘Ostrea edulis temperature’ revealed several useful papers, often Swedish research. – As the topic of ecological requirements of oysters is a very wide and complicated subject (for the authors), we just pointed out main known criteria (salinity and temperature) and added an extra reference there.

[74]: another ‘mussel’ when the authors mean ‘shell’. - corrected

[144] ‘… the need for sieving is minimal for oyster valves [1].’ This both incorrect, and the opposite of the conclusion in Reference [1]. Oysters when they are well-preserved can be as small as a thumbnail, and when they are not well-preserved the pieces of large oysters bearing the hinge (which are identifiable and countable portions of valves, left or right) can be as small as a little fingernail. Not sieving for oysters gives a massive overstatement of their average sizes, and a massive underestimate of their numbers. The lack of sieving also explains why so few other kinds of shells, especially Mytilid mussels, are found by archaeologists (line [75] of this paper). Sieving for shells including oysters is absolutely necessary, and this paper will help put Estonian archaeology ahead of the rest of Europe if it says so. – corrected

[184] ‘notches and cut marks on the interior side show that the oysters were opened while alive and eaten raw’. Shucking-marks (in North American English) only show the oyster was opened, not that it was eaten raw (the authors know this already, they explain it is only possible on Line 289). - we removed ’and eaten raw’.

[189] ‘models for tracing the exploitation of oyster-beds ...’ The main reference for using oyster shape is in fact Reference [21], so a reference to it should be included here. Oysters grade in shape from broad, frilly-edged with small hinges in shallow muddy beds to tall, boat-shaped with big hinges in offshore beds with strong currents. Most of the oysters in this paper have the shape of oysters from shallow beds in moderate currents. - Correct reference is added and the numeration of references corrected accordingly

[223] this sentence on the reservoir effect can be deleted. Almost no archaeologists will expect radiocarbon dating on strata this recent, and even fewer will expect radiocarbon dating of the shells directly. If the sentence is included, what this reservoir effect is needs explaining, which will be complicated. - The sentence deleted.

[233]: see the note above on page [189]; reference to Reference [21] needs including here. - The reference included.

[263] the sentence starting ‘However, without knowing the harvesting areas ...’ can be deleted. Nobody knows exactly which beds were harvested for archaeological oysters, the nature of the bed is always all that can be reconstructed. Striking the word ‘only’ in the next sentence ‘We can only conclude ...’ shows what the authors have done, and that they have done it well here. - These sentenses were rephrased.

[285] Most archaeological opening marks in all periods are on the ventral edge, farthest from the hinge, while modern-day opening uses a short blade from the posterior edge (the inward curve) which is the shortest distance to sever the adductor muscle: a note in this paper on how often the stab-wounds are on the rear of the shell in the modern way would help archaeologists know where and when this way of oyster-opening began. - A sentence added.

[308] another use of ‘mussel’ when ‘shell’ is meant. Here the word ‘mussel’ can be replaced by ‘flesh’ or ‘meat’. - The word ’shell’ is replaced by ’flesh’

[308]-[313]: this is the first observation of oyster-’cracking’ as an opening-method in archaeological material known to this referee, who will watch for this in his oysters and refer to this paper when he finds any. A good reason to publish this paper, editor!

[482] – [490] As an aside, the barrel of oysters from 2 Munga Street Parnu is exceptional, internationally important, and merits further research (detailed morphogmetric study as in Reference [21]; isotopic analysis to establish sea-temperature, season of harvest and provenance in Europe). – Sentence is added to show the importance of further studies.

Reviewer 3 Report

I would like to thank the authors for their research paper. I greatly enjoyed reading it. It is well written and well structured. the sample and the geographical area is clearly defined. the findings are linked to the historical events and context (conflicts, economic conditions, preferences in life-stile, etc.). The addition of visual material is a great plus. My only suggestion remaining after reading the paper is to add, if possible, an explanation as to why Baltic sea was / is not suitable for oyster cultivation.   

Author Response

The authors thank the reviewer for the comments on our manuscript. The text has been corrected accordingly:

My only suggestion remaining after reading the paper is to add, if possible, an explanation as to why Baltic sea was / is not suitable for oyster cultivation. – As the topic of ecological requirements of oysters is a very wide and complicated subject (for the authors), we just pointed out main known criteria (salinity and temperature) and added an extra reference there.